# Research-by-Design in Complex Systems: Reflections on Approaches Used to Reimagine Environmentally Sustainable, High-Welfare Poultry Housing Futures

Emma Campbell [1],*, Greg Keeffe [1], Seán Cullen [1], Anne Richmond [2], Stephen Beagan [2], Ursula Lavery [2], Brendan McKenna [3] and Steven Lester [3]

1 Architecture and Planning Department, School of Natural and Built Environment, Queen's University, David Keir Building, Stranmillis Rd, Belfast BT7 1NN, UK
2 Moy Park Ltd., Food Park, 39 Seagoe Office, Craigavon BT63 5QE, UK
3 JF McKenna, 66 Cathedral Road, Armagh BT61 8AE, UK
* Correspondence: e.j.campbell@qub.ac.uk

**Abstract:** Despite projected global rises in chicken consumption, growing environmental and welfare challenges threaten the future of commercial poultry production. Though some of these challenges, such as biosecurity, sourcing, pollution, and waste, have been thoroughly researched, the open-ended, complex, and interrelated nature of the sector means that it is difficult for poultry producers to know how to change. Design may offer a new way to analyse and reframe these challenges, to speculate on a range of different solutions for these complex systems of production. This paper reflects on the research-by-design methods applied to reimagine environmentally sustainable, high-welfare poultry housing futures. The paper is based on an eighteen-month long, multidisciplinary research project with a large U.K.-based poultry farming integrator, a poultry house ventilation and equipment supplier, and academic partners with expertise in research-by-design and bird welfare. After contextualising challenges faced by the poultry sector, the paper outlines a three-step, iterative approach within which design methods were applied, beginning with (1) a baseline analysis of farm inputs, outputs, actors, and networks, and then (2) a consolidation of themes and scenarios, leading to the development of (3) a compendium of ideas for the future of poultry farming. The Results section presents three design propositions, each imagining different futures by recreating the farm as a system of "closed-loop" flows, reframing the "chicken as client" and challenging current centralised models of production to connect consumers to food provenance and impact. These propositions function as vehicles to test design methods, such as designing for resource flows challenging actor hierarchies and hacking stakeholder networks. While some interesting ideas are presented, the paper highlights the complexity of the challenge and reflects on the value of design to reframe these challenges to collaboratively foster new perspectives and mindsets.

**Keywords:** research-by-design methods; design for environmental sustainability; high-welfare farming design; poultry housing futures; visualisation methods



## 1. Introduction

By 2030, global meat production is projected to increase by 13%, while poultry production is set to rise by 17% in the same period [1]. Although the sector is relatively low carbon compared with other livestock sectors, poultry production significantly impacts water and air quality and contributes to global resource use and waste [2,3]. In the United Kingdom (U.K.), climate legislation, including net-zero targets, requires the poultry sector to fundamentally redress operations to reduce environmental impact and build resilience to the direct and indirect impacts of climate change. Consumers are also becoming more aware of bird welfare but are reluctant to pay more for their food [1]. As such, higher welfare products still represent a small portion of the chicken market. Poultry producers

must balance these concerns along with biosecurity risks that are easier to manage through indoor housing environments. While these challenges have been explored extensively in a siloed way, their collective complexity means that it is difficult to consider how the sector might holistically adapt for the future [3]. This paper reflects on an 18-month long, applied, multidisciplinary project between the poultry sector in Northern Ireland (N.I.) and architectural design-researchers in academia to explore this question through research-by-design methods. As well as the design-research team, the project partners included Moy Park, a large U.K. (United Kingdom) poultry integrator, and JF McKenna, a poultry house ventilation and equipment supplier, and academic experts in poultry behaviour and welfare. The project was funded by Innovate UK to address and support innovation on net-zero and high-welfare agriculture through the design of future poultry housing. However, this paper is reframed towards environmental sustainability, including biosecurity, resource consumption, waste, and pollution issues.

After highlighting the challenges faced by the poultry sector and exploring typical poultry housing models, this paper reflects on the value of research-by-design methods to reimagine environmentally sustainable, high-welfare poultry house futures. It also reflects on how these methods facilitate new perspectives and mindsets within complex productive industries, in the agrifood sector and beyond. The benefit of undertaking this research in an academic setting, across an extended time-period, was in the distancing of focus from the day-to-day operations of the poultry integrator towards the exploration of long- and short-term opportunities that enable "more good" rather than attempting to simply make the poultry housing and systems "less bad". The design-research team developed an "Ideation Hourglass" framework to support top-down and bottom-up ideas, spanning different scales and timeframes to inform pathways to change. Under this framework, the team used a three-step approach to understand and reimagine poultry housing. Within each iterative, interconnected step, a range of established design tools were tested, and visualisations produced to capture, communicate, and critique ideas. The cross-disciplinary team met regularly to assess developed ideas and ideate new ones. The project culminated in the development of three propositional concepts for future poultry housing. While these propositions consolidated baseline analysis, themes, and potential scenarios developed to consider poultry sector futures, in this paper, they are used as vehicles to explore and reflect on the mix of design approaches to develop them and how these might inform ways to use design to unpack and reimagine other complex systems.

## 2. Background

### 2.1. Poultry Production Impacts and Challenges

Around 60 billion chickens are slaughtered globally per year [4]. Agriculture accounts for 14.5% of global anthropogenic greenhouse gas (GHG) emissions [5]. Poultry and egg production contributes to 8% of this figure, around 0.12% of total GHG emissions [5]. In June 2020, 1.15 billion chickens were slaughtered in the U.K., with a total production value of £2.25 billion [6]. In this context, agriculture accounts for 10% of annual greenhouse gas emissions (GHG) [7]. The poultry sector accounts for about 13% of the U.K.'s gross agricultural output and 2% of the agriculture sector's global warming potential [3]. Greenhouse gas emissions in the poultry sector are attributed to a range of operational sources. The global warming potential of standard poultry production systems in the U.K. are associated with feed and water (71.2%), electricity (3.7%), gas and oil (9.8%), housing and land (12.1%) and manure and bedding (3.2%) [3].

In its Sixth Carbon Budget, the Climate Change Committee recommended that the agricultural sector increase efficiencies in resource production, promote biodiversity, and facilitate greenhouse gas removals to address climate targets [8]. In 2022, Northern Ireland (N.I.) agreed on its first Climate Change Bill [9]. Like the UK, it seeks to meet net-zero carbon by 2050 [10]. Despite this, the challenge of making the poultry sector more environmentally sustainable is particularly acute in N.I. as agriculture is the only sector to have increased emissions since the 1990 baseline and livestock farming contributes significantly to the

regional economy [11]. The poultry sector makes up 18.5% of N.I.'s gross agricultural output, 5.5% more than the U.K. [11]. Annual greenhouse gas emissions from agriculture accounts for 26% of total emissions, 16% more than the UK [12]. Part of this can be attributed to the rapid increase in the size of the poultry sector, particularly in poultry populations for meat consumption. Between 2016 and 2020, the poultry population in N.I. rose from 14.5 to 15.4 million [13].

The rapid development of this sector has created a complex supply chain and fostered the development of supporting industries, from poultry feed manufacturers to anaerobic digestors for poultry litter management. Furthermore, land spreading is currently one of the primary ways of managing litter from broiler houses, contributing significantly to a phosphorous (P) surplus in N.I. In 2017, the national P surplus was 12.3 kg P ha$^{-1}$ [14]. Consequently, eutrophication of waterways is a prominent issue, detrimentally affecting the water quality of major rivers and lakes. In 2021, none of N.I.'s 450 river water bodies achieved good or high-level status, down from 24% in 2015, compared with 14% of English rivers rated good status [15,16]. These pressures have brought into sharp focus the environmental impact of the poultry sector and have opened questions around how best to decarbonise operations and reduce the impacts of sourcing and waste management.

Welfare is another concern for the poultry sector. Most consumers say that it is an important consideration when they purchase meat [17]. Despite this, products labelled as higher welfare, such as organic options, make up a small proportion of market share, reflecting that consumers do not want to spend more on high-welfare products. Commercial poultry producers must also balance bird welfare with biosecurity requirements, to mitigate against seasonal outbreaks of avian influenza (AI). A record number of cases of AI were confirmed in Great Britain (GB) in the winter of 2021/22 [18]. AI affects wild and commercial bird populations and presents a significant public health threat, and once detected on farms, farmers must cull tens of thousands of birds, impacting income as a result [19]. In large-scale poultry production, indoor housing is preferred, despite the impact on welfare [20]. The Better Chicken Commitment outlines ways to improve welfare in indoor-reared birds; by reducing stocking densities, careful breed selection and improved environmental standards, such as natural light, perching spaces, and good air quality [21].

### 2.2. Poultry House Typology

Historically, poultry houses were simple, lean-to timber structures capable of holding 10–12 birds. These free-range systems enabled the birds to roam extensively and required the farmer to move long distances to clean or supply feed and water to the houses. More intensive systems emerged, designed to hold hundreds, then thousands of birds at a time. These structures were timber framed, used natural ventilation, and integrated automated feed and drink lines. In recent years, conventional poultry houses have been designed as steel, portal frame structures with insulated façade panels and an uninsulated concrete floor.

Today, Moy Park's average house is approximately 20 m by 80 m and capable of holding 34,500 birds. Each house completes 6.8 cycles per year on average, so that around 234,600 birds occupy a house annually. Heating systems in Moy Park's estate use primarily biomass, with a small proportion using natural gas. As well as being artificially lit, manual opening strip windows with external shutters are installed along the length of the house to provide controlled natural light. The house is ventilated through side inlets and ridge extracts with artificial fan systems to aid air movement. This lightweight structure, with low thermal mass, is vulnerable to the increased occurrence of extreme hot and cold weather in the U.K. [22]. As a result, poultry houses rely heavily on mechanical systems to heat and move air. This is costly, wasteful, and can sometimes be insufficient at maintaining an optimum indoor environment during extreme weather events.

In recent years, notable examples of broiler and laying houses have emerged which aim to reduce operational energy use and improve bird welfare. These deviate from the conventional portal frame typology. The Rondeel is a circular laying house with alternate

day and night segments and an outer edge for play and dustbathing [23]. Even though the cost of operating the Rondeel is greater than a typical house, the design significantly improves animal welfare because it hosts smaller flock sizes and creates variable environments to support a range of needs. The Windstreek Broiler House is another example of a new poultry housing typology [24,25]. Its 11-m-high roof avails of natural cross ventilation, while heated motherhoods reduce space heating to several occupied zones. This house has a large north-facing elevation, which results in a significant increase in natural light, with glazing equivalent to 50% of the floor area compared to 3% in conventional housing. These design strategies help to reduce operational energy of the house by 80% when compared to a conventional house. While both designs offer interesting insights into the future of the poultry house, their operation is predicated on designing architectural and management solutions without explicitly integrating novel, circular systems that balance environmental and welfare concerns.

## 3. Literature and Methods

Design-research uses abductive reasoning to define complex problems and generate solutions simultaneously and across iterations in what philosopher Karl Popper defines as conjecture and refutation [26]. This is reflected in IDEO's divergent–convergent creative design process [27]. While traditional research is mostly analytical, exploring "what is", research-by-design casts into the future to explore "what if" [28]. This approach, combined with systems thinking, which supports pattern-finding in complex environments, is useful in determining solutions for "wicked" problems, which are difficult to define, incomplete and interdependent [29].

In this research study, the design-research team implicitly referred to Schön's model of Reflective Practice, which defines three stages in a design cycle: frame, move and evaluate [30]. This is mirrored in the structure of this section of the paper, which defines three steps to reimagine and visualise environmentally sustainable, welfare-friendly poultry house futures using research-by-design methods. While there are three steps described, these approaches are iterative and non-linear, with interlinks across and between.

The first step in this research-by-design approach begins with a baseline analysis of Moy Park's existing poultry housing, networks, and operations. While this step is not necessarily design-led, it highlights challenges in which the team can develop design solutions. Keeffe and Cullen describe this step as defining content in which a formal space can be contained [31]. The second step comprises a compendium of themes and scenarios to explore the wider context in which the current and future poultry house exists. The second step straddles analysis and design, to inform "what-if" questions emerging from the research in steps one and two. The third step is design-led, exploring various holistic, thematic solutions across different scales and timeframes.

To balance the mix of expertise in the project, the design-research team developed an "Ideation Hourglass" framework, shown in Figure 1, to facilitate top-down and bottom-up systems-thinking. Where Moy Park and JF McKenna brought in-depth knowledge of current systems and a focus on the incremental changes towards the project's aims, the academic team brought fresh perspectives contextualised by wider societal shifts towards radical change. The project team met regularly to discuss ideas spanning different scales and timeframes. Discussions were playful and informal, aimed at instilling cross-disciplinary ideation between the industry and academic partners to support new perspectives and mindsets about the current and future state of the poultry sector.

Within this framework, several established design methods were tested, such as the REAP method, which supports bottom-up solutions to reduce, reuse, and produce resources across different scales, and the STEEP method, which aids top-down ideation through social, technological ecological/environmental, economic, and political lens-based analysis [32–34]. During the design phase, the COCD-box method was also used to categorise ideas into how, wow, and now, aiding the team to focus on the development of just a few ideas across the spectrum of feasible, not yet feasible, common, and original [35]. Different types of

visualisations, such as diagrams and montages, were used as tools to communicate design ideas and illustrate design methods.

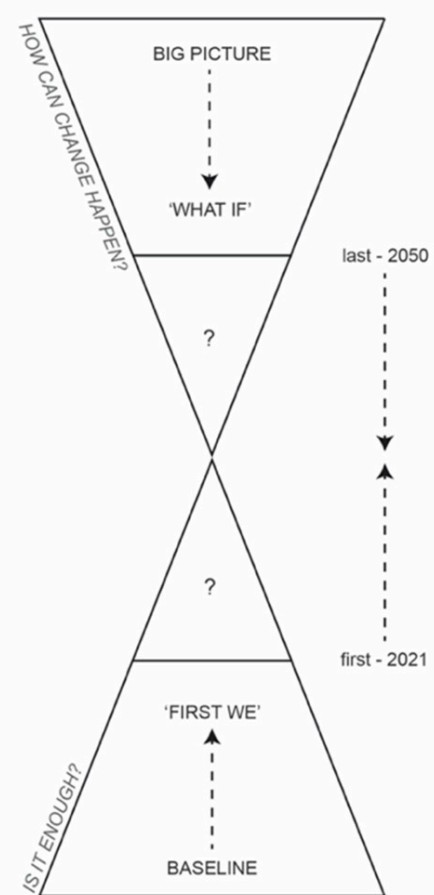

**Figure 1.** "Ideation Hourglass" framework diagram, balancing bottom-up and top-down thinking to design across timescales.

Here, the challenge of reimagining the poultry house of the future was also used to test research-by-design methods to address complex systems of production by designing spaces based on what flows through them, challenging priorities for building users by shaking up actor hierarchies and redesigning spaces based on new stakeholder dynamics and networks.

Each step in this section informs the development of design propositions, presented in the results, which also function as vehicles to reflect on the research-by-design approaches employed. Here the three design outputs are described through the design approaches used to create them.

### 3.1. Baseline of Typical Farm Inputs, Outputs, Actors and Networks

The design-research team started the project by carrying out baseline analysis of typical farm spaces and operations, including resource consumption and production. The team also analysed the actors and networks of current operational infrastructures in Moy Park.

#### 3.1.1. Assessing Embodied and Operational Carbon

Assessments of embodied and operation carbon were carried out to understand the environmental impact of poultry housing in the context of the net-zero challenge, a stipulation of the grant funding. First, the team analysed the embodied carbon associated with the construction of a typical poultry house. Through analysis of architectural drawings, and conversations with Moy Park and JF McKenna, the design-research team estimated the quantity of each construction material used in a typical house. Using the ICE embodied

carbon database [36], the quantity of embodied carbon per kg of each material type in the base (aggregate, concrete), primary structure (steel frame, timber purlins), walls (façade panels, concrete, polystyrene insulation, double-glazed windows), and roof (tin sheeting, insulation, PVC membrane) was estimated. The exercise revealed that a typical house embodies around 57.3 tonnes of $CO_2$-eq ($tCO_2$) and found that the steel structure of a typical house was particularly carbon-intensive material, using just under half of this carbon, at 25 $tCO_2$-eq per house.

The team also looked at the operational carbon required to heat and power a typical poultry house each year. Due to time-constraints, the team gathered a breadth of rough data on a typical house. For example, analysis revealed that 90% of all houses across the estate used biomass heating, and 75% of those used wood pellets; therefore, the team carried out carbon assessments based on wood pellet biomass heating. Research revealed that approximately 434,000 kWh is required to heat a house per year. The team found that houses heated by wood pellets produce 15 g $CO_2$-eq per kWh and around 6.5 tonnes of $CO_2$-eq per year [37]. Researchers found that all houses are powered by N.I.'s grid electricity, which uses around 339 g $CO_2$-eq per kWh, and each house uses approximately 34,000 kWh of electricity per year, equivalent to 10.5 homes in N.I., producing 11.5 tonnes of $CO_2$-eq per year [38,39].

Determining that half of embodied carbon in the house corresponded to its primary structure opened reflections on how changing this material to, for example, a timber structure that has lower embodied carbon, could be an easy way to dramatically reduce the embodied carbon in house construction to address net-zero targets. Since the embodied carbon of the primary structure is integral to the house, Moy Park would have to wait until an existing house reaches its end-of-life or adopt lower carbon materials in new houses first. In contrast, the team agreed that it would be easier and quicker to address operational carbon by, for example, adapting the fabric of existing houses to increase thermal mass, to insulate floors and roofs to use and waste less heat. The team also reflected on precedent typologies, mentioned in Section 2.2, to consider more efficient heating and power infrastructures and whether they could be applied in N.I. Dismantling space to focus on carbon, energy or material flows supported these reflections and unlocked new ways of designing the house to rewire these flows. The authors pick up on this point in Section 4.1 of the Section 4.

### 3.1.2. Assessing Stakeholder Actor-Networks

The design-research team unpacked and visualised other flows, from infrastructural to social, through actor-network mapping. In addition to the carbon assessment of current operations, explicit and tentative relationships between Moy Park and their stakeholders were analysed. Data were collected though informal interviews with sector specialists inside Moy Park's organisation, such as employees working on the management of litter, plastic packaging, and public relations. This informed further desktop analysis of important suppliers, consumers, influencers, and governors.

Modelled on Thun et. al's interpretation of Latour's theory, the actor-network map in Figure 2 was produced to visualise key actors and relationships between them [40]. These were used to inform conversations with Moy Park about how these networks might be manipulated to address the aims of the research project. This method of drawing helped to codify a complex and dynamic web of connections between internal and external stakeholders, providing a system lens for strategic-level conversations around present and future operations.

While Moy Park's internal business structures were clear to employees, relationships with external stakeholder networks were not. As a poultry integrator, the relationship between business and farmer is complex and intertwined, and as such, essential to understand to reimagine future relationships and spaces. One interesting aspect of the poultry integrator's business infrastructure was that the poultry farmers supplying the integrator operated a bit like franchisees, personally purchasing all the necessary ingredients from

the integrator to make fully-grown chickens, such as paying for housing, power, feed, and chicks. To determine the boundaries of environmental responsibility, the team asked the organization questions such as, Who owns the chickens? Who buys the feed? Who is responsible for litter management? Who monitors emissions? Visualisations of the whole network and key relationships in that network helped Moy Park to reflect on previously unseen connections for the first time. This opened conversations on partnerships, sharing, and responsibility on issues of resource consumption, waste, and welfare to consider how the house design might respond to or influence these concerns and highlight opportunities to hack these hierarchies to address the project's aims. This approach was taken in the development of the second design proposition presented in Section 4.2 of the Results.

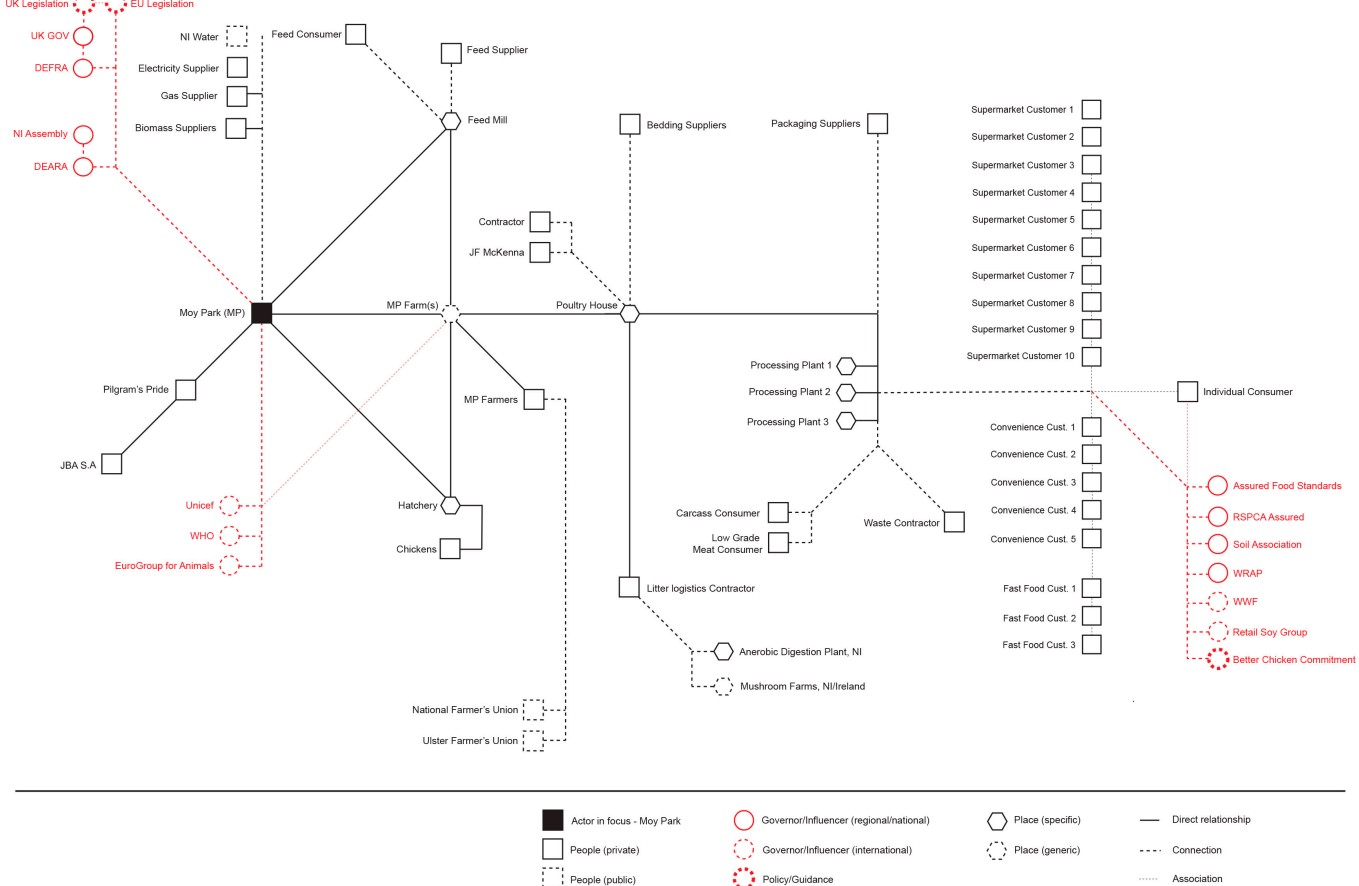

**Figure 2.** Actor-network map capturing explicit and tentative relationships within Moy Park's business ecosystem.

### 3.2. Consolidation of Themes and Scenarios

Subthemes in relation to environmental sustainability and bird welfare emerged as a result of the baseline to encourage more holistic inquiry and prompt further research to understand impacts and opportunities for Moy Park. This encouraged the industry partners to see existing conditions through new eyes allowing them to question these in a safe, explorative environment.

The design-research team developed long-term risk and opportunity scenarios using the STEEP scenario-building approach to explore what is happening in societies, including businesses, at local and global scale [32,33]. This informed the development of five themes of interest to the poultry sector. These themes, described briefly below, formed the basis of informal discussions and desktop research across the project:

1.　Biosecurity: a review of the seasonal recurrence of Avian Influenza (AI) in Europe and how it might continue to influence the scale and shape of future poultry production as well as perceptions of large-scale livestock production.
2.　Emerging technology: research on how technological advances in data management, breeding, and alternative proteins might manipulate future diets and wider consumer culture, redress stockmanship, and alter the scale of production.
3.　Animal welfare and changing tastes: assessment of recent developments and projected trends around consumer and retailer awareness and animal welfare expectations and how this might impact future operations, including house design, breeding, and economic models of production.
4.　Climate resilience: analysis of projected U.K. climate shifts and how this could impact animal welfare and supply chains, as well as the design of future houses, farms, and estates.
5.　Resource use and waste: a review of the resources consumed and produced in current poultry production operations and changing legislation affecting local and global contexts.

Theme-based analysis through scenario-building enabled the design-research team to develop a deeper understanding of core risks in achieving the aims of the project. These were communicated to Moy Park through short descriptions and icon-style diagrams to relay how production operations might be affected in the long-term. These diagrams informed a mnemonic approach, consolidating a collection of complex concerns and ideas into easily understood chunks of information. Compiling key challenges in one, easy-to-read format gave the poultry producer a discussion tool for strategic level decision-making beyond the project period. A selection of potential scenarios and mitigations is outlined in Table 1.

**Table 1.** Exploring thematic scenarios and mitigations, supported by icon diagrams.

| | Scenario | | Mitigation |
|---|---|---|---|
| Biosecurity | | - Major Avian Influenza outbreak | - Lower house densities<br>- Less birds per farm<br>- Minimum distance between farms |
| Carbon tax | | - Poultry farming must become low carbon to be economically viable | - Internalise carbon offsetting<br>- Source materials locally<br>- Reduce waste |
| Supply chains | | - Resource scarcity causes raw material price increases and volatile markets | - Close waste streams<br>- Source raw materials locally<br>- Use resources efficiently |
| Deregulation | | - Cheaper, lower welfare chicken enters market | - Revisit sustainability and welfare branding |
| Alternative Protein | | - Non-animal protein become standard in fast food and ready meals | - Increase consumer transparency<br>- Supply higher welfare chicken |

Parallels between themes revealed the complex nature of transitioning the sector to address environmental sustainability and welfare issues, as well as the impact of prioritising some themes over others, and how this might result in radically different future conditions. The research team compared complementary and conflicting themes to combine problems and solutions. These problem-solution framings are described next, ordered in relation to each of the three design propositions presented in the Results section:

- Restricting bird production to indoor conditions could ensure biosecurity by controlling conditions to reduce outbreaks. This approach also controls the flow of waste streams from the house and enables waste to be collected and repurposed to produce, for example, energy or other food products, such as mushrooms. However, an indoor-rearing approach may reduce capacity to improve animal welfare by limiting outdoor access. Conversely, reducing operational energy use and associated costs through natural ventilation opens opportunities to foster indoor–outdoor housing, which could also improve welfare but would mean ammonia emissions are less well controlled.

- Some research suggests poultry producers could rear slower-growing birds to improve welfare [41]. However, these slower-growing breeds would cost farms more because they use more heat and power and consume more feed, which has environmental and economic implications. Farmers could manage this drop in income by, for example, producing complimentary crops, while supermarkets might increase the cost of chicken to reflect increased welfare credentials.

- Emerging surveillance technologies could be employed to reduce workload and potentially address the issue of ageing farmers in the U.K. and N.I. context. Live streams of farm conditions could also be coopted by supermarket retailers to address growing welfare concerns. The poultry house might then be redesigned to become much smaller, if it is no longer entered by people, reducing construction and operational costs. However, this could have negative implications for farmers who might have to pay for the cost of technology. Welfare may also be at risk through loss of stockmanshi skills.

Compromises of one theme over another under the broader context of addressing environmental sustainability and welfare revealed different challenges for different actors within the STEEP nexus. For example, new housing infrastructures require economic investment from individual farmers, which influences the speed of change, while social and political influences could also play a role in catalysing or stalling these changes from the outside. While the STEEP analysis highlighted conflicting challenges, bearing in mind the project's aims, the research team prioritised the ecological/environmental aspect of the STEEP tool as they developed design propositions. In this way, they could employ design approaches to explore alternative systems, spaces, and networks of production unbound from the priorities of current operations. This approach of hacking stakeholder networks is explored further in Section 4.3 of the Results.

### 3.3. Imagining a Compendium of Multi-Scalar Ideas for the Poultry Farm of the Future

The design-research team used the themes, scenarios, and baseline findings to build a compendium of propositional ideas and solutions to imagine environmentally sustainable, higher welfare poultry farms and housing. The compendium of future scenarios, spaces, and ideas developed organically, often evolving from unstructured conversations. In these instances, problem-framing through "what-if" questions led to thought-experiments: *"If this happened, then what might be the result? How might this impact society, the environment, or economics? If we removed economic considerations, what decisions might be made, and could the environment or society benefit more?"*. From here, the team applied Cross' concept of a creative leap, finding "sub-solutions" to bridge against the framed problem through conceptual thinking [42]. Significant ideas stemming from conversations were recorded and then were informally framed as a research question. This question was then explored by one team member through more rigorous design-based investigations and presented to the design team on a weekly basis for feedback to inform new iterations. This approach

worked well, giving time to collaboratively bounce ideas through conversation as well as ensuring progress by formalising one or two of these ideas alone. Working in this way, each individual design-researcher brought different value systems, interests, knowledge profiles, and experiences to their propositional outputs. Presenting these formalised ideas in a group setting challenged these value systems, providing positive criticism through different lenses. This meant that one idea, explored by several individuals, would result in different outputs, and therefore more holistic solutions.

Through the "Ideation Hourglass" framework, the project team balanced top-down and bottom-up actions. Where the top-down approach focused on risks and opportunities as well as external pressures, the bottom-up approach focused on immediate action to identify "low-hanging fruit", or operations that could be easily transitioned to environmentally sustainable or higher welfare alternatives. Aided by the baseline analysis findings, the design-research team used the REAP design method to categorise potential "easy wins" at the house, farm, and estate scale in Figure 3 [34]. Taking a mindset that it is better to fix a little bit of the whole system than perfect just one aspect of it, one change applied at each scale could have a multiplier effect to quickly reduce carbon emissions, waste, or reliance on external supply chains.

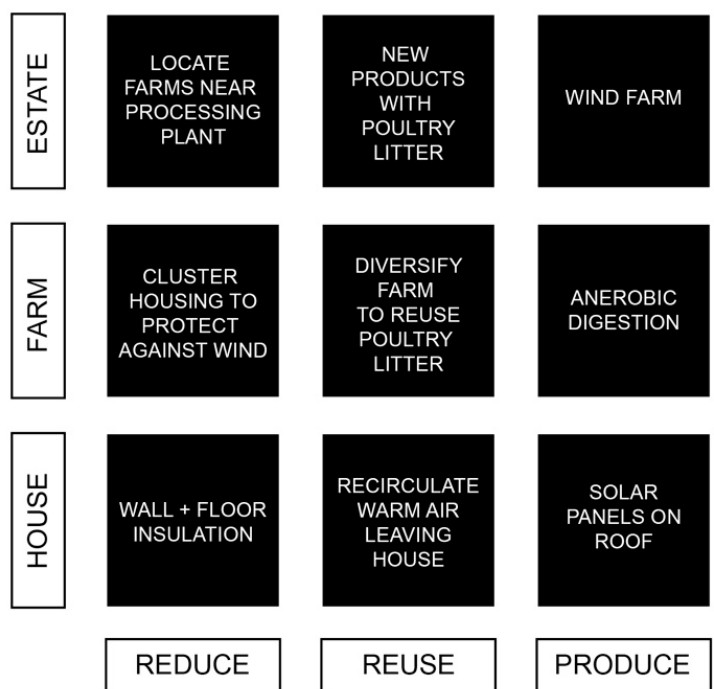

**Figure 3.** REAP method matrix for categorising bottom-up design ideation.

As research-by-design is iterative, some propositional ideas emerged earlier in the project than others, for example, reviewing typical annual energy use to power a poultry house prompted testing of the potential energy production if solar panels were to be installed onto its roof. Equally, some ideas were developed further than others, while others remained abstract or focused on a certain aspect of the project's aims. In addition, as is typical in design-research, evaluative thinking ensured that problems and solutions co-evolved across the project. For example, development of the propositional ideas incited further baseline analysis as well as further theme definition and scenario-building [26]. Themes placed at the centre of the divergent–convergent thinking process funneled the development of further ideas, while also helping the design-research team to focus on developing just a few ideas more comprehensively. The COCD box method was also used to facilitate critique and design decisions to develop the three propositions [35].

In all instances, each relevant idea was recorded and communicated through a range of drawing formats including diagrams, maps, timelines, collages, orthographic draw-

ings, and sketches with one example provided in Figure 4. The design-research team applied a familiar visualisation approach that combines the production of diagrams and photomontage-style collages to communicate core design ideas and give a sense of how these ideas play out spatially [43]. Read together, both drawing types communicate complex ideas holistically. Read in context, they capture views of an imagined scenario combining existing and future worlds. Drawing across the formats described prompted on-demand decision making: "*How big is the future house in its landscape? How does light enter the space? What type of food is the chicken consuming?*". Once complete, the image supports further reflection on the type of space imagined and whether it effectively addresses the challenges. Like the scenarios described in the previous section, this approach worked as another visual mnemonic to explore many solutions to various parts of a complex design problem. Further examples of the different drawings produced are shown in the Results section of this paper.

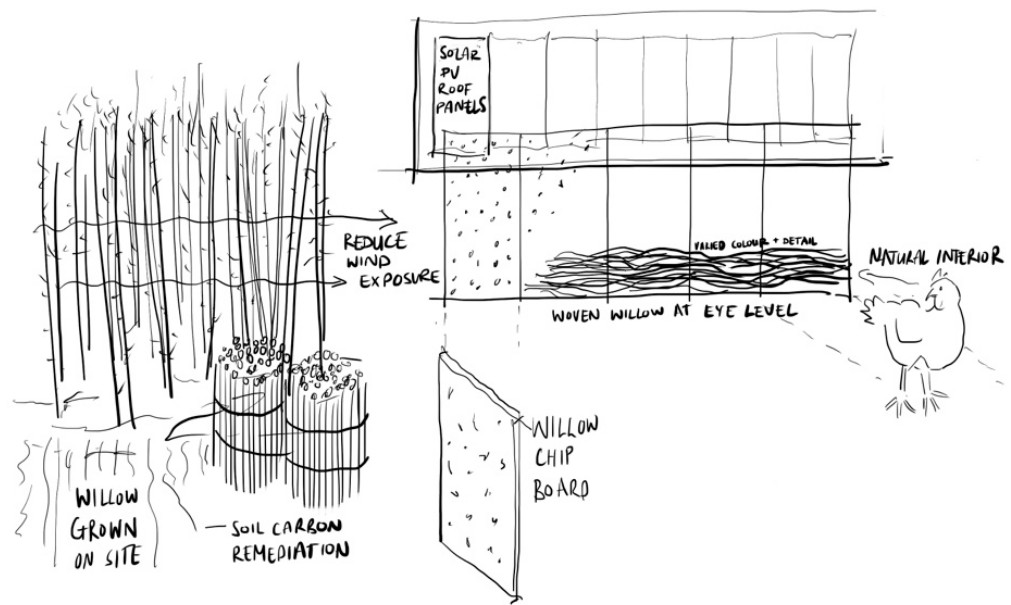

**Figure 4.** Sketches depicting early ideas on embedding circularity within the poultry house design and wider farm operations.

## 4. Results

The research-by-design methods applied across this project aimed to generate a way of looking at current and reimagining future poultry house architectures to address environmental sustainability and bird welfare in poultry production. Here, the challenges faced by the poultry sector are tested, with the overarching aim of finding design approaches to examine and unlock pathways to reimagine complex locked-in systems of production.

As indicated in the previous section of this paper, the last step in the approach informed the development of a compendium of design propositions to address these aims. Following the generation of five key themes in Section 3.2, a compendium of disparate thematic design ideas was synthesised to the design and visualisation of just three propositional futures for poultry production, explored in this section. Each proposition emerged from lens-based investigation of two of the five themes identified in Section 3.2, as indicated in Figure 5, to inform house designs with different priorities relating to the overarching challenges.

The propositions, named *Circular Chicken, Happy Chicken,* and *Network Chicken*, are described below—less as finished projects or optimum visions for the future of poultry housing, and more as vehicles to describe the different research-by-design methods employed in the project to address its overall aims. These propositions are described under the umbrella of the design approaches employed in the project, for example, the *Circu-*

*lar Chicken* proposition highlights how the designers shifted their focus from space to flow design.

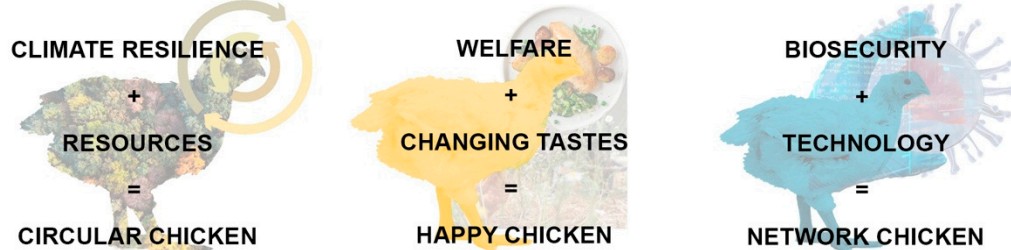

**Figure 5.** Icon images indicating how different themes, described in Section 3.2, were combined to generate different scenarios for the future of poultry farming and housing.

### 4.1. Dismantling Space to Flows

Baseline analysis of the typical poultry house, highlighted in Section 3.1.1, revealed the carbon emissions associated with construction materials, heating, and power supply. This revealed that the constant stream of resources used to operate the house outweighed the one-time impact of constructing the architectural artefact, and as a result, dismantled the initial view that, to achieve net-zero, the house must first be redesigned. This highlighted the need to redesign the house's flows rather than its spaces.

The difficulty of reducing emissions associated with operating the house since, for example, Moy Park is unable to decarbonise external heat and power supply, was also highlighted. This opened conversations around how they might use existing waste streams within their estate to localise heat and power supply, for example, through anerobic digestion of poultry litter. The team also discussed how Moy Park might take advantage of the large roofscape on existing poultry houses to install solar PV panels to build some resilience against the rising price of heat and power supply. The team reviewed the types of technologies required to support this shift, such as battery storage, and how Moy Park might share excess energy production through cooperatives with residents in rural N.I. Further baseline research on the impact of waste outputs, such as poultry litter, revealed unseen environmental damage, which informed conversations around how to promote more sustainable resource supply, use, and management in line with UN (United Nations) Sustainability Goal 12, Responsible Consumption and Production [44]. This shifted priorities away from the initial focus on net-zero, laid out by the funding body, to address environmental sustainability in a more holistic way. All of the above were ingredients in the development of *Circular Chicken* proposition.

Abstracting typical poultry farm operations into a system of resource flows, shown in Figure 6, the *Circular Chicken* poultry house was reframed as a container for inputs and outputs from the perspective of material flows, such as water, to nutrient and chemical flows, such as ammonia. The proposition explored ways to valorise poultry litter, taking ammonia and waste heat from poultry litter for aquaponic production of high value crops. It looked at an existing waste stream from anerobic digestion, a liquid called digestate, and how this might be used to produce micro-algae to feed insects and localise poultry feed or make biomethane for the poultry integrators fleet. It also looked at ways to make mycelium packaging from mushrooms grown in poultry litter and how urban forestry might be used to localise bedding supply. This proposition not only revealed the economic value of existing waste streams but also expanded the houses' boundaries, reconnecting the scale and impact of resource use and waste with global ecologies.

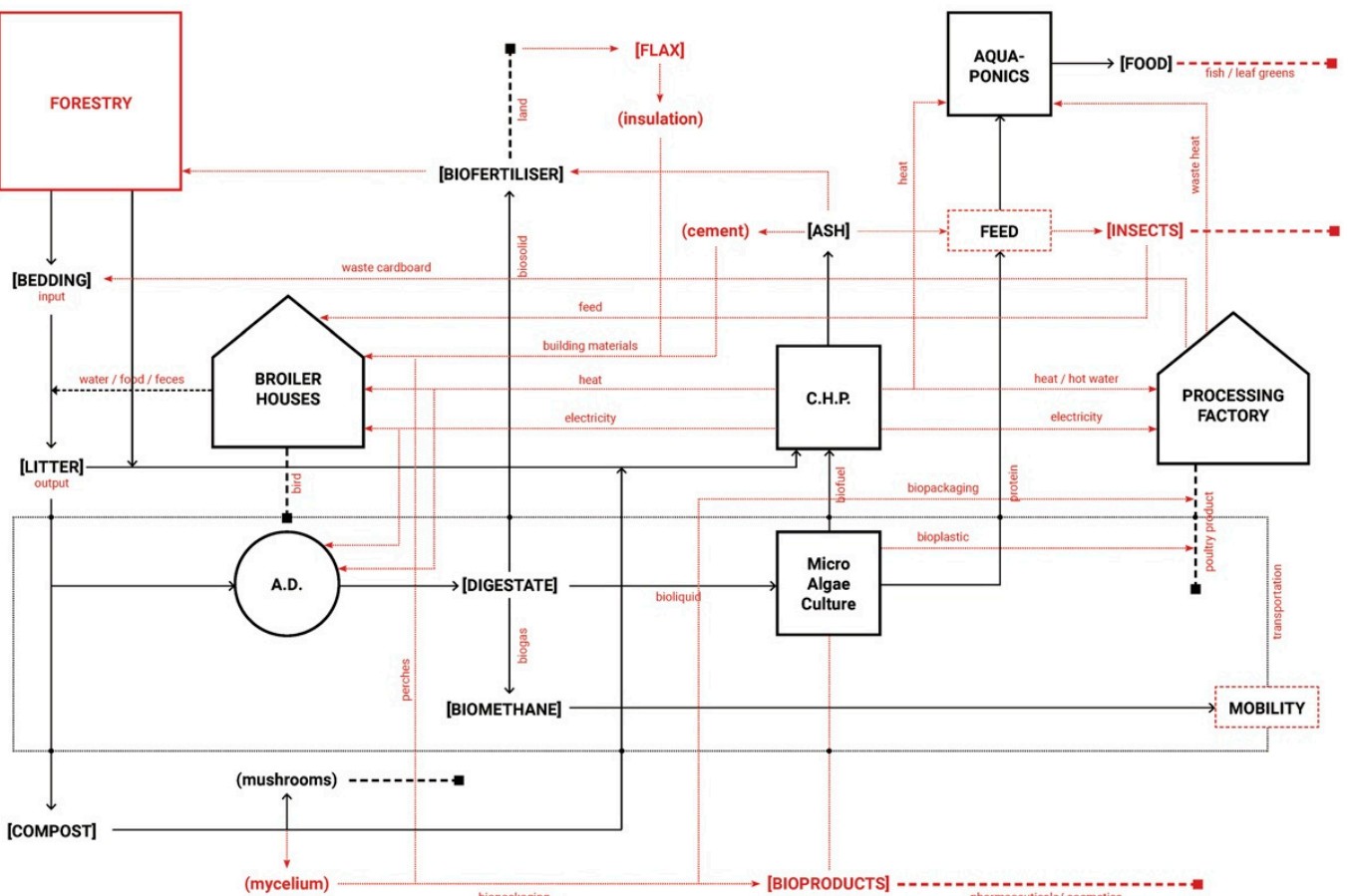

**Figure 6.** Diagram exploring potential material flows to reduce waste and create new economic opportunities. Research for this diagram was funded separately by CIEL to Moy Park.

Reflecting on the existing linear model of poultry production, the design-research team applied circular economy principles to reimagine the future poultry farm as a system of closed loops. The proposal was first described through abstract flow diagrams and sketches. Then through collage images, shown in Figures 7 and 8, visualising the interventions required to facilitate waste-free on-farm production, such as an offset forest and micro combined heat and power plant (CHP). Discussed together, they highlighted the value of designing whole-system interventions that go beyond the redesign of spaces or the installation of technologies. The visualisations opened conversations around the scale of these interventions, whether it is possible to, for example, have an anerobic digestor on each farm or whether one needs to be available to multiple farms in an area to make this option more economically viable.

Viewing productive businesses such as this one as a global resource manipulator, the proposition also opened questions about how Moy Park should respond to current and future climate legislation around carbon accounting, and how the design of poultry housing might help the farm, rather than the business estate, to become an accounting boundary, halting the flow of waste streams beyond this boundary and optimising flows to create new spin-off income streams, such as through the production of mushrooms or algae while targeting existing issues of environmental pollution, such as eutrophication.

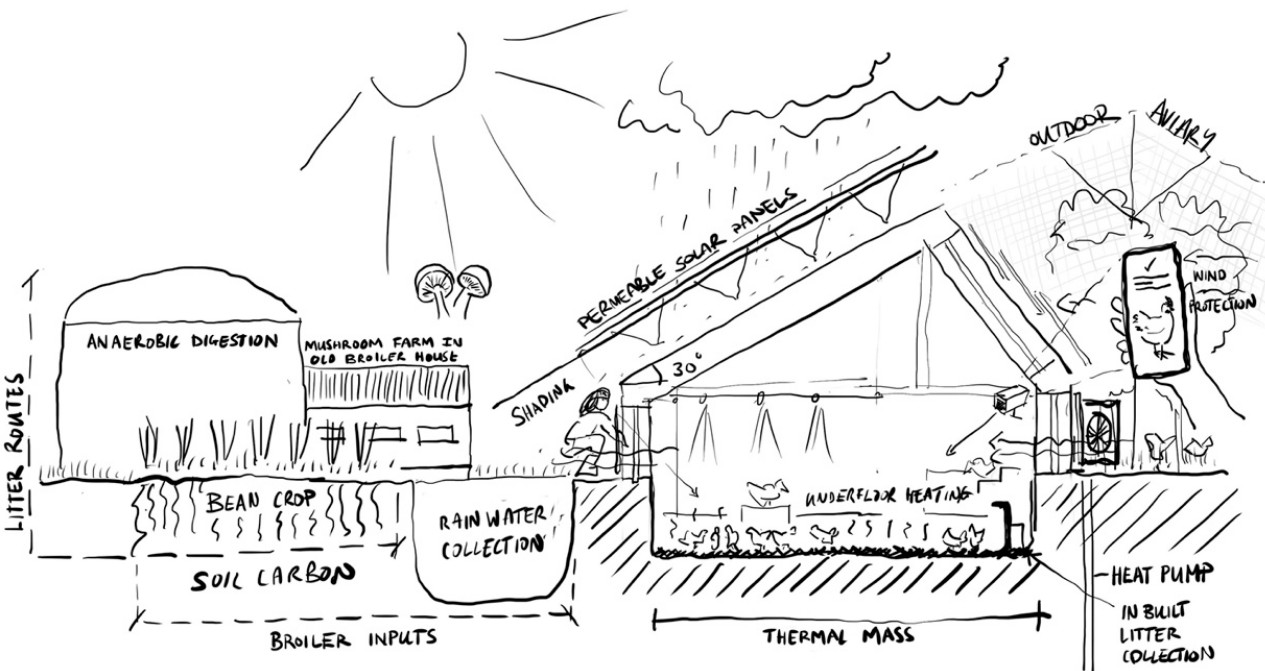

**Figure 7.** Sketch section exploring relationships between potential architectural, infrastructural, operational, and technical components on a future "circular" poultry farm.

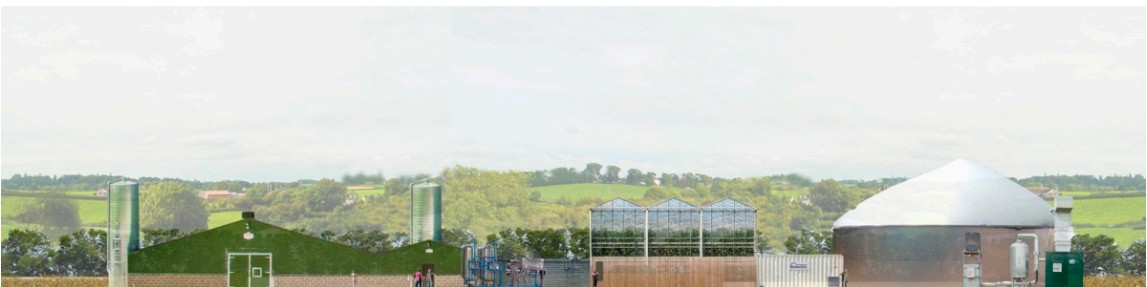

**Figure 8.** Elevational collage depicting what a "circular" farm might look like and contain.

### 4.2. Restructuring Actor Hierarchy

Spatial analysis of the evolution of Moy Park's housing revealed that bird welfare is addressed incrementally within the existing large-scale, tech-centric housing model to ensure high levels of biosecurity. Working closely with poultry behaviour experts, the design-research team developed the *Happy Chicken* proposition, shifting priorities towards welfare and climate resilience, two of the five themes in Section 3.2.

The team played on common architectural practice approaches by developing a client brief for a chicken, which functioned as a tool to give a voice to a silent, non-human actor in the productive system. This playful approach to engagement and brief-making temporarily re-orbited dynamics away from business priorities, providing a flattened hierarchy to expand conversations beyond the current state-of-play. This approach set a mission for the design-research team, animal behaviour, and welfare experts as well as Moy Park and JF McKenna to come together to understand and design for a chicken's dietary, lifestyle, and environmental preferences. Numerous discussions with experts and farmers revealed that, like humans, chickens have different personalities, for example, some are more active and playful than others. Spatial analysis of several types of internal, external, and hybrid housing systems enabled further conversation around designing for chicken happiness. This research was visualised through a series of chicken profiles detailing, for example, how chickens play, what they eat, and daily routines, shown in Figures 9 and 10.

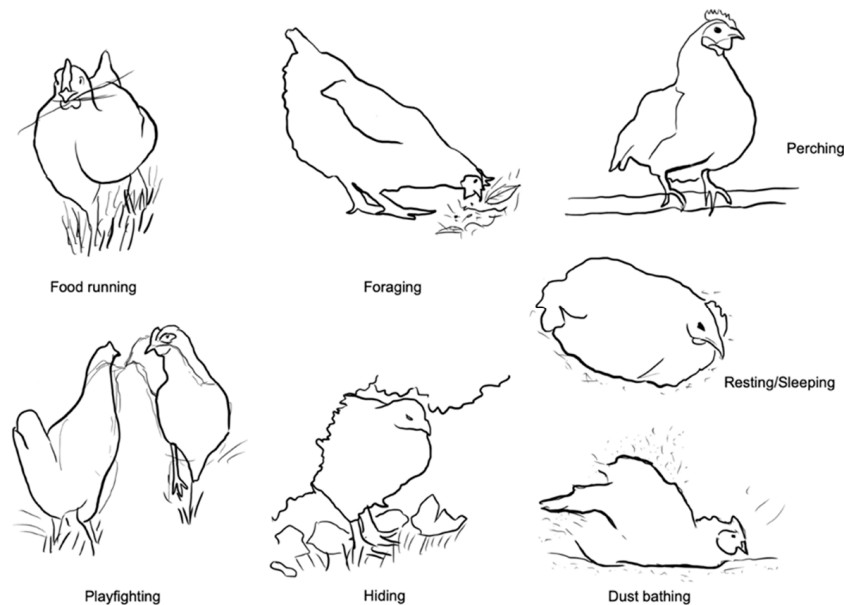

**Figure 9.** Sketches developed with poultry welfare experts to facilitate conversations around internal poultry house design ideas.

| Name | Age | About me | Hobbies | Favourite memory | Favourite time of day | When I'm most scared |
|---|---|---|---|---|---|---|
| Jo | 37 days | Broiler chicken | Food running Dust bathing | When I jumped on top of straw bales | When I take a stroll in the morning | When the humans caught my friend |

**Figure 10.** Sample bird persona developed with poultry welfare experts to facilitate conversations around bird behaviours and preferences across age and breed.

Illustrated in Figure 11, initial ideas focused on creating an integrated forest housing system to resituate more resilient, slower-growing chicken breeds in their indigenous environment, support the immediate offsetting of carbon emissions, and diversify farm income through short coppice crop production. Inspired by organic practices, a time-based design strategy imagined that the future poultry house could be situated within a new forest of oak standards, infilled with hazel coppice. These houses would be constructed from local timber and designed to move through the forest at the end of each production cycle. Flexible netting pulled between trees would allow the chickens to be in a protected outside space within nature. The movement of houses across the forest would be timed to coincide with coppicing and sequenced to fertilise the ground appropriately, while at the same time managing the risk of contamination between flocks.

Retaining the chicken as a key actor allowed the design-research team to question existing processes and scales of production. The proposal opened debate about how designing for bird happiness, through nature-based solutions, may incur higher economic costs. The proposition highlights the biosecurity risk of allowing chickens to live outdoors and seeks to offset this with smaller flocks of more resilient breeds. At the same time, the proposition presented opportunities to diversify income, improve supply security, and support recognisable consumer branding as well as rapid carbon descent through reforestation. Attaching a mission to the design strategy through discussion-based fact-finding and profile-building proved effective in helping the poultry integrator to remain open to seeing the value of solutions that were different from current modes of production. Shaking up actor hierarchies opened opportunities to see Moy Park's operations from

a previously unconsidered perspective, even if this approach was deemed economically unviable for current scales of production.

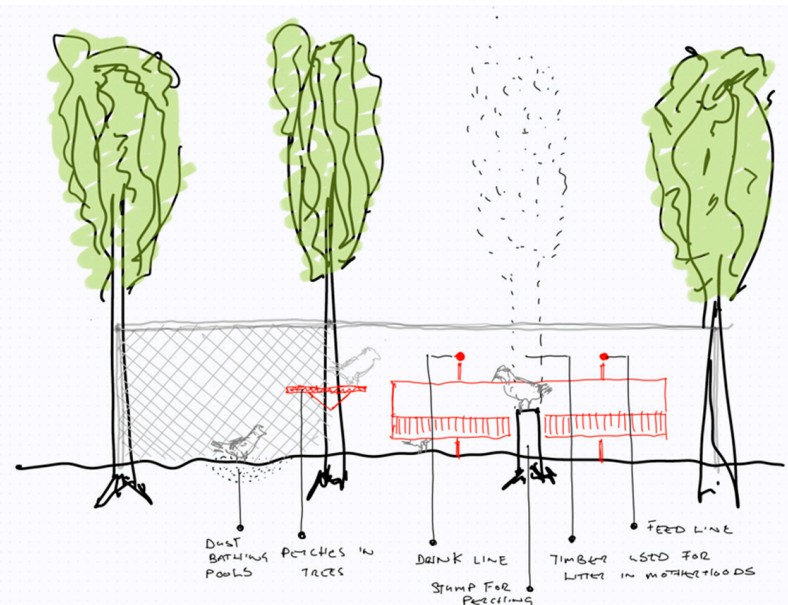

**Figure 11.** Early sketch proposal for a moveable house in a forested environment that repositions the chicken as the primary client.

### 4.3. Hacking Stakeholder Networks

Researching U.K. supermarket targets, Moy Park's key customer, revealed plans to increase the number of sustainable, higher welfare meat products as well as plant-based products in stores. For example, Tesco aims to source all soy from deforestation-free regions by 2025, and in the same period, plans to increase sales of plant-based alternatives by 300% [45]. Similarly, research by LIDL suggests that 71% of their shoppers want retailers to be clearer about how the chicken they purchase was raised and 68% of 16–24-year-olds are drawn to plant-based diets [46]. Unpacking these targets created a framework to understand emerging and future expectations around poultry production. Stakeholder mapping, described in Section 3.1.2, supported this notion of the supermarket as the frontier for market access. It also highlighted the complex economic relationship between the integrator and individual farmer, and how this could support or hinder rapid business transformation.

Parallel conversations across the study focused on the importance of maintaining biosecurity and the integrator expressed interest in maintaining a centralised, indoor housing environment to address this concern. Reflecting on the changing consumer-food dynamic, the design-research team sought to reframe this "red line" to imagine a smaller, distributed model for poultry farming, based on creating outbreak exclusion zones to reduce contamination risks and smaller flocks to reduce economic risks if disease outbreaks occur. The *Network Chicken* proposition, visualised in Figure 12 emerged from this concept of designing small, seen, and secure housing. Challenging current centralised models, the concept supports better connections between consumers and their food's provenance. Suitable for urban, domestic sites, the consumer becomes the farmer and custodian of the flock. Managed through artificial intelligence (AI), fed with domestic food waste, monitored by roaming "expert" farmers and visited by a mobile abattoir; the house, or coop, shrinks the gap between consumers and food to enable transparency and visibility in the food-chain. This proposition addresses environmental sustainability through the use of food waste, reducing the need to source feed globally. The proposition raises questions around how this model could be managed, who would own the chickens, and how much would they cost.

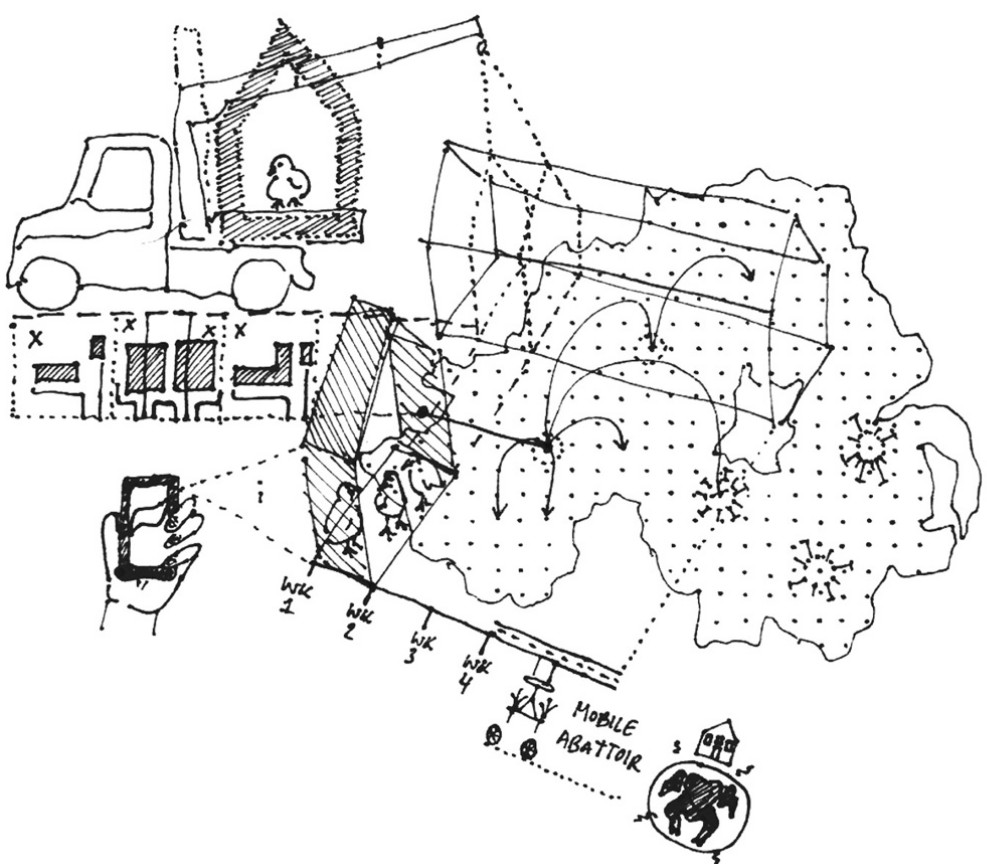

**Figure 12.** Early concept sketch proposal for a distributed network of small houses across N.I, monitored and managed by consumers using app technology.

Rather than think of this proposal as an artefact, the design-research team combined current and future priorities of several stakeholders, such as Moy Park's desire for biosecurity and the consumer's desire for transparency. Using icon-style diagrams such as those shown in Figure 13, the team communicated a mix of external influences potentially impacting poultry production. Amplifying these priorities, the *Network Chicken* proposition is extreme and tied to significant cultural and technological leaps. Despite this, it served well to help the wider research team to reflect on how internal and external stakeholders might shape the future of food production and consumption, develop an awareness of the type of emerging technologies likely to catalyse shifts, and explore how current business operations help or hinder rapid changes.

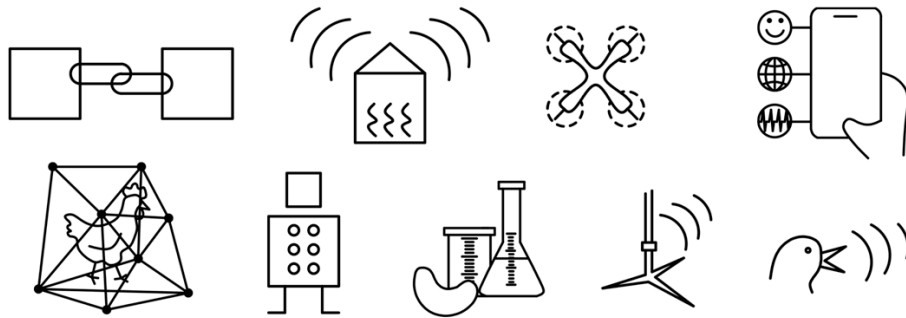

**Figure 13.** Icon diagrams produced as a mnemonic to catalogue and communicate shifts in consumer and technology trends. Top row, left to right: blockchain, weather sensing, drones, off-site management, Internet of things, robotics, alternative protein, RFID-tagging, welfare sound sensing.

## 5. Discussion and Conclusions

### 5.1. Reflections on the Design Methods Used

Across the project, the design-research team used systems thinking to find pathways from problems to solutions to address the challenge of designing a more environmentally sustainable, higher welfare future poultry house, recognising the "wicked" and complex nature of the challenge. Reflecting on established design methods, the team structured the design process into three iterative, interlinked steps which, starting with baseline analysis of farm inputs, outputs, actors, and networks, then developed themes and scenarios and finally a compendium of solutions. Within each step, a range of design methods, including STEEP, REAP, and the COCD box, were deployed to develop and test ideas. An "Ideation Hourglass" framework supported the development of top-down and bottom-up solutions. Spanning the long and short-term, this framework bridged complementary and conflicting concerns to gather a range of ideas from across the interdisciplinary design team. Different types of visualisations were used as devices to communicate and critique ideas in project team meetings. Diagrams were particularly useful in communicating complex challenges and ideas in an accessible format. Visualisations also flexibly communicated solutions for both near and far-future queries.

Applying different lenses of investigation to the house led to a variety of scenarios and three design propositions exploring the future of poultry housing. Named *Circular Chicken, Happy Chicken,* and *Network Chicken*, each proposed radically different futures arriving from an exploration of different themes and scenarios and reflections on the baseline findings. Each proposition also tested different design methods. The *Circular Chicken* proposition emerged by shifting the focus away from the design of space to the design of flows. Reflecting on the baseline carbon assessment of housing, which found that the carbon embodied in a house with a 30-year lifespan is much less than the carbon to heat and power housing annually, this design proposed integration between house, farm, and estate to enable resource efficiencies through circular economy principles. The *Happy Chicken* proposition restructured actor hierarchies in the poultry production system to design the poultry house for its primary occupants, and then speculated on ways to make this economically and operationally viable. The final proposition, *Network Chicken*, is a reflection of the stakeholder mapping carried out in the baseline analysis. It explores how external actors and technologies might enable decentralised models of production that are bio-secure and better connected to consumers.

Rather than attempt to realistically determine one future poultry house, the propositions tested design methods to explore the future of specific, emerging trends already impacting the poultry production sector. Analysing the poultry house by what flows through it, how it connects to actors-networks and could be impacted by societal shifts revealed that a poultry house is not just a piece of real estate; it is also a home for chickens, a consumer of biomass, a producer of protein, and a node along a supermarket supply chain. By zooming in on the poultry house through different lens, the design approaches taken offered a route to recognise the complexity and interdependency of the challenge yet quickly generate spatial solutions to support changed mindsets that start to break away from locked-in systems. This approach revealed the difficulty of the challenge and that there is no right or clear answer for how the future poultry house should be designed, particularly in the context of climate change and consumer shifts. Though not explored in the scope of this paper, Moy Park carried out sensitivity testing and an internal economic assessment at the end of the project to inform investment strategies. Some of the ideas arising from this project informed contingency planning and strategic visioning while others spurred further in-house research activities on specific environmental sustainability and welfare challenges.

### 5.2. Designing in a Changing Context

Undertaking this study through the COVID-19 pandemic and during the Brexit transition period highlighted the speculative nature of planning for the future within an uncertain

present. Global events such as this revealed how quickly some trends accelerate while others dissipate or change. Throughout the project, the concerns of climate change and the vulnerability of the sector to global supply shocks became immediate and apparent to all partners, particularly the poultry integrator. These events included Northern Ireland recording its hottest day on record during the summer of 2021, a shortage of key workers and drivers, reoccurring AI outbreaks, increasing energy prices and a shortage of $CO_2$, all critical to the poultry sector. In addition, the UN's 26th Climate Change Conference highlighted rapidly changing global and local legislation shifts towards environmentally sustainable, high-welfare food production [34,35]. These pressures exposed the fragility of current practices and processes, highlighting the reliance on linear supply chains and tangential industries that are politically, economically, and environmentally volatile.

*5.3. Fostering Interdisciplinary Collaboration*

While the research did reveal some interesting ideas and trajectories for poultry production and consumption, one clear benefit of the design methods used was that it supported industry partners to form new perspectives about their sector's future, though aided somewhat by the timing of external events during the project. The design approaches taken helped Moy Park and JF McKenna crash-test pathways to change, shifting from incremental to rapid, radical shifts and identify ways to make "more good" in how Moy Park's operates, rather than tweaking existing systems to make things "less bad" [33]. In a sector that is continually in production, time and effort tends to be spent dealing with problems in the immediate term. This project provided industry partners with the space to consider the day to day in the context of bigger future challenges.

The design-research team tested ways to foster collaborative ideation with industry partners during team meetings. For example, at the beginning of the project, the team started by explored far-future what-ifs rather than unpacking the existing architecture. This approach dismantled the wider team hierarchy bringing everyone together to see problems associated with current production practices by first thinking of big-picture solutions. Discussions remained intentionally informal and playful, often peppered with silly questions challenging the status quo to invite collaborative reflection and build consensus. Together with the methods, this design environment enabled industry partners to "see the forest for the trees", to play out different scenarios without risk, co-create better models based on different value systems, and to map pathways to implement these in future operations.

*5.4. Limitations to the Research-by-Design Methods Used*

The spectrum of opportunities that design can unlock—the possible versus the probable—also presents a limitation. The method, shaped by questions of "what" or "who" is placed at the centre of the challenge and is fundamentally determined by the designer's ethical and theoretical position. For example, placing the chicken at the centre of the brief produces a radically different solution to one that places a technology at the centre of the challenge. It is not to say that one approach is better than the other, but that the full spectrum of opportunities can never fully be realised because of the bias that underpins the approach of those involved in the design process. Furthermore, the design methods used throughout this research work require time and space to explore around and beyond the immediate question at hand. In industry, limited time and resources mean that this is not always possible. As such, ensuring mechanisms for exploring the capabilities offered by analysing long-term opportunities is vital in industry and academia. Equally, industry partners must be open and transparent with information and actively engage in the ideation process. Naturally, this openness can lead to tough questions and even tougher solutions.

*5.5. Future Work*

This paper highlights the value of research-by-design to reframe, reimagine, and visualise solutions addressing interdependent problems within the poultry sector. The design-research team used an explorative approach, testing a mix of design methods and

visualisation techniques within a framework that spanned different scales and timeframes. Informal, playful discussions within the project team fostered ideation between academic and industry partners, bridging top-down and bottom-up action. Further work could expand on the baseline analysis findings or explore further propositional ideas about the poultry house, based on new combinations of themes and scenarios. More work is also required to understand how the design approaches explored in the Results, such as dismantling spaces to flows, might be optimised or compared against one another. Finally, this paper also highlights interesting insights into the value of design-led collaboration between academic and industry partners to consider the future of complex systems.

**Author Contributions:** Conceptualization, E.C., G.K., S.C., A.R., S.B., U.L., B.M. and S.L.; Methodology, E.C., G.K. and S.C.; Validation, E.C., G.K., S.C., A.R., S.B., U.L., B.M. and S.L.; Formal analysis, E.C. and S.B.; Investigation, E.C. and S.C.; Resources, A.R., S.B., U.L., B.M. and S.L.; Data curation, E.C., G.K., S.C. and S.B.; Writing—original draft, E.C., G.K. and S.C.; Writing—review & editing, E.C., G.K., S.C., A.R., S.B., U.L., B.M. and S.L.; Visualization, E.C. and S.C.; Supervision, G.K., S.C., A.R., U.L. and B.M.; Project administration, G.K., S.C., A.R., S.B., U.L., B.M. and S.L.; Funding acquisition, G.K., S.C., A.R., S.B., U.L. and B.M. All authors have read and agreed to the published version of the manuscript.

**Funding:** This research was funded by Innovate UK, grant number 50015.

**Institutional Review Board Statement:** Not applicable.

**Informed Consent Statement:** Not applicable.

**Data Availability Statement:** Not applicable.

**Acknowledgments:** The authors would like to express sincere thanks to Niamh O'Connell for their invaluable knowledge and insight on bird welfare. Complementary research on poultry litter valorisation carried out by the design-research team alongside this project was funded by CIEL. The authors recognize Theano Stoikidou for their input in this separate study which fed into the design process.

**Conflicts of Interest:** The authors declare no conflict of interest.

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
