# Peer review of "Research-by-Design in Complex Systems: Reflections on Approaches Used to Reimagine Environmentally Sustainable, High-Welfare Poultry Housing Futures"

_sustainability, doi:10.3390/su15075808_

Round 1

Reviewer 1 Report

1) First of all, following the suggestions of Stewart1, this reviewer considers that we are not talking about methodology but "method" or a set of "methods". Looking at row 47, it is clear to this reviewer that it should read "methods" rather than "methodologies". The same happens in row 393: "The methodology applied across this project..." should really read "The method applied across this project..." This looks like a minor thing but it is not. Scientists must demonstrate a full domain of the scientific language.

2) The title "A Climate and Welfare Transition Design Methodology Applied to the Poultry Sector" supposes that it is specifically designed to assess "climate and welfare transition". As stated in rows 129-131, it seems more like a mix of design-research approaches applied to the development and spacial visualization of a climate transition strategy, in this specific case, to the poultry sector. We may say a good application indeed, adapted but not specifically designed for it.

3) When presenting poultry production in the UK, the authors provide some statistics which, isolated, do not mean too much. Saying that agriculture accounts for 10% of annual GHG emissions or that that account for 26% in the case of Northern Irelands means nothing if you don´t put them in a global context. That the CCC says that addressing GHG emissions from agriculture is key if the UK is to achieve net-zero targets. Why should the UK achieve net zero? If climate change is a global issue, what would be the impact of this? Is it worth it? Who is willing to pay for it? In the same section, the authors also put some attention to pollution issues, which have nothing to do with climate change. Eutrophication is of course an important issue and, in this case, is a local issue, not a global one. Thus, addressing this environmental problem is likely to be more beneficial, economically and socially, to the British consumer.

4) In section 3, the authors present the characteristics and features of design-research approaches. While correct, even architectural design research approaches are basically general, adequate to but not specific to draw climate transition strategies. Back again to the carbon discussion, saying that a typical house embodied around 57.3 tons of CO2, half of which is due to the steel structure, does not provide very useful information either. The same comment is valid for the tons of C derived from operational activities (heating and powering). Moreover, with regard to the strategy for easy visualization of carbon impact, while undoubtedly better for communication to not specialized audiences, may also be meaningless. What really means that forest typically sequesters 2.9 tons/ha of carbon dioxide? What really means that a typical house has around 20ha of forest or 0.65 across a 30-year lifespan? It does not add any meaningful information without putting it in its due context. The same comments are valid, in the opinion of this reviewer, with regard to the numbers exposed in the case of carbon used to heat and power a typical house.

5) Looking at rows 214 to 216, one reads "The team also agree that going off-grid in terms of power would have significantly more impact across Moy Park's estate than, for example, making the house more efficient". Under what grounds they drew that conclusion? Going off-grid and what? Going to wind or solar energy? As it is perfectly known that power generation from both sources is not permanent, what happens when they are not generating? In rows 2016-218, the authors claim that "providing energy storage is integrated into any future off-grid energy system, this would not only reduce environmental impact but also potentially reduce the cost of electricity and ensure supply at lows costs." Although recognizing implicitly that the big problem today is energy storage, this is a declarative statement without any real support from evidence. In fact, the authors do not cite any. Of course, all of us have the belief that sooner or later, the human brain will solve this problem and will create new technology to overcome these issues but there is no clear sign of it. Clearly, solar and wind sources are still distant from being "clean". To the non-constant power generation problem that is inherent to both technologies, one must add some other issues, such as the carbon footprint of building solar panels and windmills, the "visual pollution" caused by their implementation at large scales (there are gruesome examples), as well as the "environmental pollution" problems they will represent in the near future at the end of their lifespan and must be discarded.

6) In rows 268 to 271 of the manuscript one may read: "The design-research team developed long-term and opportunity scenarios using the STEEP scenario-building methodology" (again it should say "method"). However, this reviewer did not find the specific results of this step, especially the "economic" leg of this method. In many parts of this manuscript, one may find references to "economy", "economic", and so. However, there is no economic analysis at all nor any real reference to that. The manuscript talks about "understanding of core risks in achieving a climate transition in the poultry sector" (rows 292-293) and the impacts of "prioritizing some themes over others, and how this might result in radically different future conditions" (rows 295-297). But they do not provide any explanation or results in that sense. The authors talk about "propositional ideas" (row 336) and explained that "some ideas were developed further than others, while others remained abstract or focused on a certain aspect of the project´s aims" (rows 339-340). All seem to be closer to proposing some methodological approach to examine complex problems, rather than generating credible and feasible results from the application of the methods to address potential issues in the poultry industry.

7) With regard to the results, the authors expressed that the project aimed to generate a way of looking at current and future poultry house architectures to address climate change and bird welfare Rows 393-395). However, the discussion seemed to go beyond that specific point. The authors continue by stating that "the last step in the methodology informed the development of a compendium of design propositions to address this aim." According to the authors, "this compendium was synthesized to the design and visualization of just three propositional futures for poultry production." (397-398). From this point, the authors describe the three propositions (circular chicken, happy chicken, and network chicken), which seems to represent (from the first to the third, in that order) a supposedly greater commitment to environmental issues without any discussion about the economic and social implications.

8) On the other hand, the authors point oy that "Research on projected consumer trends revealed a sustained shift towards plant-based diets and increased awareness and concern around the environmental and welfare impacts of meat consumption" (517-519). The authors rely on this assertion, which may be certainly true, without any critical vision if this concern is supported by science or not. For instance, the assessment that plant-based diets are better than meat diets for the environment and welfare impacts is, at least debatable, if we take into account all the factors involved (ie, the additional land that must be devoted to crop production, as well as fertilizers and other, the increment of the risk of erosion, soil structure, increase in the use of energy, etc.). Of course, it is not the objective of this paper to assess that issues but something should be mentioned for an honest discussion, not taking some things for granted. The discussion lacks the necessary equilibrium in the discussion as the not-specialized reader may be induced to think that all plant-based diets are better than meat-based ones for the environment, even when these ideas are popular and reflected in UK supermarket targets.

9) In the second paragraph in section 4.3 "Hacking Stakeholder Networks", it reads "parallel conversations across the study focused on the importance of maintaining biosecurity in future iterations of the net-zero, high-welfare poultry house and the integrator expressed interest in maintaining a centralized, indoor housing environment to address this concern" (rows 530-533). Biosecurity is essential, always. So essential that it is hard to believe that it could be an issue with the current level of development of the British poultry industry. At least at the level of being the building concept of the Network Chicken alternative.

10) On page 16, at the beginning of the discussion chapter, the authors recognize that "the designed outcomes are exaggerated but do not fully detach from reality, supported throughout by already emerging trends" (rows 572-574). Moreover, in the following sentence, they added that "rather than try to predict the future, these imaginary settings and spaces support exploration around the future of specific, emerging trends already impacting the poultry production sector" (574-576). If this were clearly stated as the aim of the work, it would be more acceptable for this reviewer. Expressions like "outcomes are exaggerated but do not fully detach from reality" and "these imaginary settings and spaces" look more appropriate for a "what if exercise". This reviewer also believes that the questions posed in rows 580-583 are not fully adequate. How can we say that a proposed future path is more desirable without fully discussing all the "pros and cons", including of course the economic ones? The results presented in the manuscript do not provide any scientific argument about this. The statement: "This approach revealed the complexity of the challenge and that there is no right or clear answer for how the future poultry house should be designed, particularly in the context of changing climates" (595-597). This must be the main highlight or conclusion of the whole work. As a rational agent, Moy Park "highlighted their desire for near-market solutions and required help to decide how they should invest in technology" (605-606). Saying that "the design researchers found they brought more value in exploring future technologies in a broader context through theme-building and far-future scenarios" (608-609) does not mean, by any means, that a "broader context" can exclude a thorough economic analysis. In order to accept increasing costs without a clear estimation of the profits, they would ask for guarantees that no one else can offer the same product at a lower cost. Without an effective and prior monopoly power, it will imply market regulations where the extra costs are paid by somebody, in general, the consumers.

11) In connection with the previous point, the authors draw a conclusion that sounds more like an implicit elicitation from the discussion than a real conclusion supported by the evidence provided by the research. In rows 692-694, the authors say that "the design-research team has come to a conclusion that may seem counter intuitive-integrate more. Integration, not just within the poultry sector but in tangential ones too". First, if this is not a conclusion drawn by the authors (they say that the design-research team did so), then it should be placed under the discussion to avoid misleading the reader. If it pretends to be considered as a conclusion of the paper, this reviewer does not understand under which grounds it may be so. Talking about integration in poultry or any other industry is not such a trivial thing as can be stated in a two-line sentence and without further explanation about its connotations and impacts. It requires a solid understanding of some topics, such as industrial organization, and market fundamentals, among other things, all within the area of economics.

Finally, this author recognizes the effort and time devoted by the research team in presenting this manuscript. Although the idea and the topic are of interest, this reviewer honestly believes that the outcome is not sufficient for its publication in its current form. I suggest a major revision of the title, objectives, and results that really emerge from the application of these methods. On the same line, the conclusions have to flow effectively from the results, avoiding statements that, even when the authors believe are true, do not clearly emerge from the presented evidence.

1 Stewart, I.M.T. (1993) Reasoning and method in Economics; an introduction to Economic Methodology. Gregg Revivals. Brookfield, Vermont (USA).

Author Response

Thank you for taking the time to provide a really comprehensive review of our paper, it is much appreciated. Your comments have helped the authors to reflect on the submission and make changes that they believe significantly improve the quality of the research presented. Answers to your comments are provided below:

  1. First of all, following the suggestions of Stewart1, this reviewer considers that we are not talking about methodology but "method" or a set of "methods". Looking at row 47, it is clear to this reviewer that it should read "methods" rather than "methodologies". The same happens in row 393: "The methodology applied across this project..." should really read "The method applied across this project..." This looks like a minor thing but it is not. Scientists must demonstrate a full domain of the scientific language.

Agree. Thank you for the reference. The authors have adjusted the language in our manuscript, where appropriate.

  1. The title "A Climate and Welfare Transition Design Methodology Applied to the Poultry Sector" supposes that it is specifically designed to assess "climate and welfare transition". As stated in rows 129-131, it seems more like a mix of design-research approaches applied to the development and spacial visualization of a climate transition strategy, in this specific case, to the poultry sector. We may say a good application indeed, adapted but not specifically designed for it.

Agree. The authors have adjusted the title of the manuscript and reframed its contents to focus on the research-by-design methods applied to reimagine poultry housing. The paper no longer focuses on transition design or climate change, instead, it reflects on the types of design methods used to understand the complexity of the challenges facing the poultry sector. In response to your comment and those from other reviewers, the two overarching challenges, noted in the title, are now [1] high welfare and [2] environmental sustainability, which includes issues of resource use and waste, including pollution.

3.Saying that agriculture accounts for 10% of annual GHG emissions or that that account for 26% in the case of Northern Irelands means nothing if you don´t put them in a global context.

Agree. We have added additional references to compare UK and NI data and contextualise them with global data. See rows 111-112, 155-422.

Why should the UK achieve net zero? If climate change is a global issue, what would be the impact of this? Is it worth it? Who is willing to pay for it?

Partly agree. This paper accepts the fact that the UK has set net-zero targets. Questions of why net-zero targets have been set and the general implications of net-zero, such as who is willing to pay for it and whether it is worth it, are beyond the scope of this manuscript. Instead, its focus is on using design methods to unpack and ideate solutions for future poultry housing to address environmental sustainability and higher bird welfare in the sector. In response to this comment, we have added a few sentences recognising the project funder, Innovate UK, as well as the original ambition of the project, which was to address net zero in the poultry sector and how the paper is reframed to consider broader environmental issues beyond net-zero. See rows 129-132.

In the same section, the authors also put some attention to pollution issues, which have nothing to do with climate change. Eutrophication is of course an important issue and, in this case, is a local issue, not a global one. Thus, addressing this environmental problem is likely to be more beneficial, economically and socially, to the British consumer.

Partly agree. As mentioned in our answer to question 2, we have adjusted the title and framing of the manuscript, shifting the focus from climate change to environmental sustainability, within which pollution issues are considered. Due to this change, eutrophication is worth noting as it is a result of pollution linked to poultry production. However, whether eutrophication is a global or local issue is not the focus of this paper. Likewise, who benefits from addressing agricultural pollution is also not in this paper’s focus. It is important to mention pollution and its impacts because it has negative environmental implications and designers can reflect on this as they reimagine new types of poultry houses. For example, the author uses the issue of pollution as the basis to explore the Circular Chicken proposition, which aims to optimise resource consumption and eliminate waste streams, such as land-spreading which is linked to eutrophication (row 1814).

4 In section 3, the authors present the characteristics and features of design-research approaches. While correct, even architectural design research approaches are basically general, adequate to but not specific to draw climate transition strategies. Back again to the carbon discussion, saying that a typical house embodied around 57.3 tons of CO2, half of which is due to the steel structure, does not provide very useful information either. The same comment is valid for the tons of C derived from operational activities (heating and powering). Moreover, with regard to the strategy for easy visualization of carbon impact, while undoubtedly better for communication to not specialized audiences, may also be meaningless. What really means that forest typically sequesters 2.9 tons/ha of carbon dioxide? What really means that a typical house has around 20ha of forest or 0.65 across a 30-year lifespan? It does not add any meaningful information without putting it in its due context. The same comments are valid, in the opinion of this reviewer, with regard to the numbers exposed in the case of carbon used to heat and power a typical house.

Partly agree. The authors have removed reference to ‘architectural’ research-by-design approaches and instead highlighted in row 125 that the design-research team are from an architecture discipline. The carbon assessment was carried out in the context of the net-zero challenges, a stipulation of the grant funding, now noted in the text in rows 129-132 and 709-711. Assessing embodied and operational carbon opens questions about how to design the poultry house of the future, and the types of interventions that can be implemented sooner, such as insulating floors, or in the longer term, such as changing the primary structure material (rows 733-746). The calculations are retained because they may be helpful comparisons for industry or academic researchers in the agri-food sector as they carry out their own carbon assessments. Comments from several reviewers regarding the forest visualisation approach have been really enlightening and interesting. The authors have agreed to remove the forest visualisations from this paper and develop a separate publication which gives appropriate space to interrogate and discuss this aspect of the research.

5 Looking at rows 214 to 216, one reads "The team also agree that going off-grid in terms of power would have significantly more impact across Moy Park's estate than, for example, making the house more efficient". Under what grounds they drew that conclusion? Going off-grid and what? Going to wind or solar energy? As it is perfectly known that power generation from both sources is not permanent, what happens when they are not generating? In rows 2016-218, the authors claim that "providing energy storage is integrated into any future off-grid energy system, this would not only reduce environmental impact but also potentially reduce the cost of electricity and ensure supply at lows costs." Although recognizing implicitly that the big problem today is energy storage, this is a declarative statement without any real support from evidence. In fact, the authors do not cite any. Of course, all of us have the belief that sooner or later, the human brain will solve this problem and will create new technology to overcome these issues but there is no clear sign of it. Clearly, solar and wind sources are still distant from being "clean". To the non-constant power generation problem that is inherent to both technologies, one must add some other issues, such as the carbon footprint of building solar panels and windmills, the "visual pollution" caused by their implementation at large scales (there are gruesome examples), as well as the "environmental pollution" problems they will represent in the near future at the end of their lifespan and must be discarded.

Partly agree. The authors have removed references to going ‘off-grid’, which were originally in section 3.1.1. References to energy production and storage are relocated to section 4.1. Dismantling Space to Flows, in the Results. This section explores the design proposition, Circular Chicken, through the lens of designing flows rather than spaces. The framing of this section is all about reflecting on the real design conversations that took place, in relation to how to design with flows. Rows 1721-1736 in this section, refer to solar PV panels, battery storage, and cooperatives as some of the design considerations in the brief-making process. Including these references capture how the Circular Chicken design proposition developed. Although ’visual pollution” and “environmental pollution” caused by renewable technology waste are valid considerations, they did not feature in the development of this particular design proposition.

  1. In rows 268 to 271 of the manuscript one may read: "The design-research team developed long-term and opportunity scenarios using the STEEP scenario-building methodology" (again it should say "method"). However, this reviewer did not find the specific results of this step, especially the "economic" leg of this method. In many parts of this manuscript, one may find references to "economy", "economic", and so. However, there is no economic analysis at all nor any real reference to that. The manuscript talks about "understanding of core risks in achieving a climate transition in the poultry sector" (rows 292-293) and the impacts of "prioritizing some themes over others, and how this might result in radically different future conditions" (rows 295-297). But they do not provide any explanation or results in that sense. The authors talk about "propositional ideas" (row 336) and explained that "some ideas were developed further than others, while others remained abstract or focused on a certain aspect of the project´s aims" (rows 339-340). All seem to be closer to proposing some methodological approach to examine complex problems, rather than generating credible and feasible results from the application of the methods to address potential issues in the poultry industry.

Partly Agree.

  1. As stated in earlier responses, the authors have reframed this paper as a reflection on the design methods used in the project and no longer seek to provide ‘credible and feasible’ results in relation to poultry sector challenges. This is highlighted across the paper, see examples in rows 133-137,149-152, 681-689, 1614-1620, 2096-2098, 2102-2107.
  2. The beginning of the Literature, Materials and Methods section, now sets the scene for the design-centric methods applied. In this section, the STEEP method is highlighted, alongside others used in the project (row 672-676). The STEEP method is mentioned again within 3.2 Consolidation of Themes and Scenarios, in rows 1035-1039 and 1295-1304.
  3. The authors have added additional references to economic considerations across the paper (see rows: 1262, 1306, 1323,1746, 1759, 1863, 1931, 1962, 1970, 2092, 2108) and Table 1 (pg. 9), however also clarify that the overarching focus of the paper was on environmental sustainability and high-welfare (see rows: 20, 135, 596, 1320-1321, 1343-1344, 2064-2065, 2126). Therefore, in the context of STEEP, there was more weighting on the first E in the acronym (environmental/ecological) (see rows: 1304-1313).
  4. Some of the propositions did have economic considerations. In reviewing flows rather than spaces in the Circular Chicken proposition, the design-research team revealed the economic potential of waste streams such as poultry litter. Restructuring actor hierarchy to design houses for chicken welfare, in the Happy Chicken proposition, the design-team also explored ways in which the farm might diversify crop production or how the supermarket might rebrand the product to make the outcome more economical.
  5. The authors expand on the ‘core risks’ by including a table highlighting potential scenarios and mitigations developed during the project (Table 1). Further discussion on conflicting and complementary themes is provided in rows 1249-1303, where each of the three bullet point sections set the scene for how each propositional idea in the results emerged.
  6. As the paper focuses on research-by-design methods, economic analysis is outside its scope. However, this was carried out internally by Moy Park at the end of the project to inform investment strategies (rows 2107-2109).

7 With regard to the results, the authors expressed that the project aimed to generate a way of looking at current and future poultry house architectures to address climate change and bird welfare Rows 393-395). However, the discussion seemed to go beyond that specific point. The authors continue by stating that "the last step in the methodology informed the development of a compendium of design propositions to address this aim." According to the authors, "this compendium was synthesized to the design and visualization of just three propositional futures for poultry production." (397-398). From this point, the authors describe the three propositions (circular chicken, happy chicken, and network chicken), which seems to represent (from the first to the third, in that order) a supposedly greater commitment to environmental issues without any discussion about the economic and social implications.

Partly Agree. Reflecting on the previous answer, specifically 6C and 6D, economic considerations are embedded in the brief discussion in developing the propositions however environmental and ecological considerations have been prioritised to address the wider aim of the project. Beyond this, the authors have clarified across the paper that the propositions are vehicles to test different design methods (rows 150, 687, 1616, 2078), therefore the results are more about this than developing the right solutions for the poultry sector, noted in 2090-2101 of the conclusion.

8 On the other hand, the authors point oy that "Research on projected consumer trends revealed a sustained shift towards plant-based diets and increased awareness and concern around the environmental and welfare impacts of meat consumption" (517-519). The authors rely on this assertion, which may be certainly true, without any critical vision if this concern is supported by science or not. For instance, the assessment that plant-based diets are better than meat diets for the environment and welfare impacts is, at least debatable, if we take into account all the factors involved (ie, the additional land that must be devoted to crop production, as well as fertilizers and other, the increment of the risk of erosion, soil structure, increase in the use of energy, etc.). Of course, it is not the objective of this paper to assess that issues but something should be mentioned for an honest discussion, not taking some things for granted. The discussion lacks the necessary equilibrium in the discussion as the not-specialized reader may be induced to think that all plant-based diets are better than meat-based ones for the environment, even when these ideas are popular and reflected in UK supermarket targets.

Partly agree. As Moy Park’s main consumers, supermarkets have made plans to offer more plant-based products in store. Whether plant or meat-based diets are good or bad for the environment or welfare is not a concern for the authors and lies outside the scope of the paper. The statement illustrates a threat to Moy Park's future operations and must be considered as they plan future operations. For balance, the authors note that supermarkets intend to sell more sustainable, higher-welfare meat products alongside plant-based products (rows 1947-1949).

9 In the second paragraph in section 4.3 "Hacking Stakeholder Networks", it reads "parallel conversations across the study focused on the importance of maintaining biosecurity in future iterations of the net-zero, high-welfare poultry house and the integrator expressed interest in maintaining a centralized, indoor housing environment to address this concern" (rows 530-533). Biosecurity is essential, always. So essential that it is hard to believe that it could be an issue with the current level of development of the British poultry industry. At least at the level of being the building concept of the Network Chicken alternative.

Partly Agree. Biosecurity is a huge concern for poultry producers and threatened public health in the U.K. The authors agree that this was not made clear enough in the original submission. Biosecurity issues in the poultry sector are now mentioned in 1. Introduction (rows 119-121), and discussed in 2.1 Poultry Production Impacts and Challenges (rows 426-433)

10 On page 16, at the beginning of the discussion chapter, the authors recognize that "the designed outcomes are exaggerated but do not fully detach from reality, supported throughout by already emerging trends" (rows 572-574). Moreover, in the following sentence, they added that "rather than try to predict the future, these imaginary settings and spaces support exploration around the future of specific, emerging trends already impacting the poultry production sector" (574-576). If this were clearly stated as the aim of the work, it would be more acceptable for this reviewer. Expressions like "outcomes are exaggerated but do not fully detach from reality" and "these imaginary settings and spaces" look more appropriate for a "what if exercise". This reviewer also believes that the questions posed in rows 580-583 are not fully adequate. How can we say that a proposed future path is more desirable without fully discussing all the "pros and cons", including of course the economic ones? The results presented in the manuscript do not provide any scientific argument about this. The statement: "This approach revealed the complexity of the challenge and that there is no right or clear answer for how the future poultry house should be designed, particularly in the context of changing climates" (595-597). This must be the main highlight or conclusion of the whole work. As a rational agent, Moy Park "highlighted their desire for near-market solutions and required help to decide how they should invest in technology" (605-606). Saying that "the design researchers found they brought more value in exploring future technologies in a broader context through theme-building and far-future scenarios" (608-609) does not mean, by any means, that a "broader context" can exclude a thorough economic analysis. In order to accept increasing costs without a clear estimation of the profits, they would ask for guarantees that no one else can offer the same product at a lower cost. Without an effective and prior monopoly power, it will imply market regulations where the extra costs are paid by somebody, in general, the consumers.

Party agree. Section 1. Introduction and 5. Discussion and Conclusion are amended to reflect the aims of the paper, to reflect on the research-by-design methods used in the project. The ‘what if’ nature of research-by-design approaches is also clarified at the beginning of section 3. Literature and Methods (rows 585-592). The authors agree that the statement: "This approach revealed the complexity of the challenge and that there is no right or clear answer for how the future poultry house should be designed, particularly in the context of changing climates", is an appropriate conclusion for the paper in relation to speculating the future of poultry housing. As stated in our answer 6F, Moy Park undertook an internal economic assessment after the project, however, the outcomes are not included within the scope of this paper. The authors have added further discussion on the collaborative, cross-disciplinary aspects of the project, captured in the top-down and bottom-up approach taken (rows 658-666, 2129-2147)

11) In connection with the previous point, the authors draw a conclusion that sounds more like an implicit elicitation from the discussion than a real conclusion supported by the evidence provided by the research. In rows 692-694, the authors say that "the design-research team has come to a conclusion that may seem counter intuitive-integrate more. Integration, not just within the poultry sector but in tangential ones too". First, if this is not a conclusion drawn by the authors (they say that the design-research team did so), then it should be placed under the discussion to avoid misleading the reader. If it pretends to be considered as a conclusion of the paper, this reviewer does not understand under which grounds it may be so. Talking about integration in poultry or any other industry is not such a trivial thing as can be stated in a two-line sentence and without further explanation about its connotations and impacts. It requires a solid understanding of some topics, such as industrial organization, and market fundamentals, among other things, all within the area of economics.

Agree. The authors have revised the conclusion so that it more implicitly relates to the results and the overall aims of the manuscript.

Reviewer 2 Report

Dear authors,

despite the high originality of the paper, there are certain issues concerning the research methods used in this paper and how methods and results have been presented.

There is no mention of the funding received for this study, while there is no information of any ethical committee approval for deducting this research.

In addition, the chicken-centric approach proposed here is presented in rather hilarious approach, that does not fit with the character of Sustainability Journal.

if the authors wish to resubmit the papers, major revisions including principles of research and redesigning the methodology, revising the tone of the paper in a more elevated/scientific approach.

Author Response

Thank you for taking the time to review our submission.

This paper employed design methods and has been submitted to the Design and Sustainability special issue. The new title and updated contents echo this. Now, the paper is a reflection on the research-by-design methods used to reimagine environmentally sustainable, high-welfare poultry futures. In section 3. Literature and Methods, the design methods build on existing approaches and are clearly described and well-referenced. The results are also design-led. The propositions function as vehicles to explore the different design methods employed by the architectural design-research team to consider the future of poultry housing. In the Circular Chicken proposition, for example, the team design flows rather than spaces to develop solutions that are more resource efficient.

The funding body was mentioned in the back matter of the submission, which doesn’t form part of the submission to reviewers. However, the authors agree that it is useful to include the funding body within the manuscript and have updated this accordingly (rows 129-132). Equally, notes on ethical committees are normally contained in the back matter. In this project, an ethical committee was not required.

The authors disagree with the comments made about the ‘chicken-centric’ approach described in section 4.2 Restructuring Actor Hierarchy and noted in section 5. Discussion and Conclusion. By restructuring actor hierarchy, this design approach helped the project team to holistically consider ways to improve bird welfare, by making chicken the ‘client’ for the future poultry house. Indeed, this is a playful approach, but play is an integral ingredient to design ideation which helps unlock fixed perspectives and mindsets. Through scenario-building and ‘what-if’ narratives, design teams can interrogate the priorities of different actors, in this instance, consumers, farmers, poultry producers and chickens, to reconsider a new balance between these priorities to radically improve bird welfare.

Reviewer 3 Report

This manuscript addresses a relevant issue regarding the transition of the poultry sector to a climate neutral and chicken friendly environment. This is an important topic in the international agenda, as the poultry sector represents an important share in terms of GHG emissions. A holistic approach was conducted to assess the sustainability of poultry farming in Northern Ireland. The goals to be addressed are adequately framed and the topics are presented in a transparent and comprehensive way. A design-research methodology was adopted in a three-step approach aiming to identify long-term opportunities and risks for the poultry sector and predict the poultry farm of the future. The adopted methodology enabled the effective involvement of all the players, with fruitful discussions on the pros and cons of critical issues. Furthermore, the methodology may be applied to other sectors. Finally, the manuscript is well-structured and includes a comprehensive list of references. These are the main strengths of the manuscript. Nonetheless, I include a few comments (in order of appearance) which I think may improve the overall quality of the submitted document:

1)  Lines 36-37: I expected the sum to be 100%;

2)  Line 190: ref [22] calculates an average of 2.2-2.9 tonnes of CO2. Why do the authors use the top of the range? Furthermore, did the authors check if there is any inconsistency between tC and tCO2 (a multiplier of 3.67 applies when converting mass of carbon to mass of carbon dioxide);

3)  Line 191: it is not clear for the reader if this is an annual figure;

4)  Lines 203, 204: What do the authors mean with “This source used…”? Are these CO2 emissions from the combustion of wood pellets? If so, this CO2 has been previously sequestered from the atmosphere. As such, is it accounted similarly to CO2 emissions from the UK electricity mix (probably from a fossil energy source?

5)  Line 207: Using the authors’ figures, I calculate 3.97 ha per house…

6)  Lines 214-219: These judgements require further analysis for the sake of clarity. Why going off-grid have significantly more impact but going off-grid with energy storage reduces the environmental impact? Please elaborate a little further on this issue;

7)  Lines 251-255: The manuscript would benefit if additional data on how these relationships evolved was provided (e.g. flowcharts, tables, etc.);

8)  Lines 293-295 & 304-309: Additional info concerning this issue would benefit the manuscript, e.g. showing the trade-offs between themes;

9)  Line 301: “accelerative” should read “accelerate”;

10) Figure 4: I wonder if the second theme in the “Happy Chicken” scenario should be “Climate Resilience” instead of “Changing Tastes”;

11)   Line 413: section 3.1.2 does not include any data on supply/delivery of resources/products; 

12)   Line 445: “… farms in in a…” should read “… farms in a…"

13) Lines 576-584 & 629-636: Following previous points, I missed a more systematic approach and presentation of the different alternatives/scenarios addressed in this research as well as the relationships between queries and proposed solutions.

Author Response

Thank you for taking the time to review our submission, your comments have been very helpful. Useful questions raised by you and one other reviewer in relation to the forest visualisations have helped the reviewers to decide to remove this from this paper and develop a separate publication that can give enough space to reflect on this part of the project. Other answers to your comments are provided below:

  1. Lines 36-37: I expected the sum to be 100%;

Thank you, this was a simple miscalculation (rows 395-397)

  1. Line 190: ref [22] calculates an average of 2.2-2.9 tonnes of CO2. Why do the authors use the top of the range? Furthermore, did the authors check if there is any inconsistency between tC and tCO2 (a multiplier of 3.67 applies when converting mass of carbon to mass of carbon dioxide);

This question is now out of the paper’s scope. The authors have doubled checked calculations that remain.

  1. Line 191: it is not clear for the reader if this is an annual figure;

This question is now out of the paper’s scope.

  1. Lines 203, 204: What do the authors mean with “This source used…”? Are these CO2 emissions from the combustion of wood pellets? If so, this CO2 has been previously sequestered from the atmosphere. As such, is it accounted similarly to CO2 emissions from the UK electricity mix (probably from a fossil energy source?

This should read ‘this source produced’ and is in relation to emissions once burned (see rows 727-729). Please, see reference here: https://www.gov.uk/government/publications/greenhouse-gas-reporting-conversion-factors-2021

Line 207: Using the authors’ figures, I calculate 3.97 ha per house…

This question is now out of paper's scope.

  1. Lines 214-219: These judgements require further analysis for the sake of clarity. Why going off-grid have significantly more impact but going off-grid with energy storage reduces the environmental impact? Please elaborate a little further on this issue;

The authors have removed references to going ‘off-grid’, which were originally in section 3.1.1. References to energy production and storage are relocated to section 4.1. in the Results. This section explores the design proposition, Circular Chicken, through the lens of designing flows rather than spaces. The framing of this section is all about reflecting on the real design conversations that took place, in relation to how to design with flows. Rows  1737-1759 in this section, refer to solar PV panels, battery storage, and cooperatives as some of the design considerations in the brief-making process. Including these references capture how the Circular Chicken design proposition developed. It is worth also noting that the authors have reframed the paper throughout, and amended the title, to highlight the paper's focus is to reflect on the design methods used by the design-research team. Locating some of the project team discussions around energy use and production within the results are used to highlight the design brief-making process.

  1. Lines 251-255: The manuscript would benefit if additional data on how these relationships evolved was provided (e.g. flowcharts, tables, etc.);

This is a great point, thank you. The authors have included one diagram of the actor-network relationships between Moy Park and the poultry sector see Figure 2. Diagrams produced during the project to describe the relationship between Moy Park and farmers contain sensitive information therefore cannot be included in this paper.

  1. Lines 293-295 & 304-309: Additional info concerning this issue would benefit the manuscript, e.g. showing the trade-offs between themes;

A new table (1) (pg 9) has been added to the method section to illustrate some of the scenarios and mitigations discussed within the project. The authors have added further discussion points in rows 1243-1303, to set the scene for how the propositions developed and the types of trades-offs required in the process. Rows 1304-1313 highlight these trade-offs further, where ultimately the team prioritised environmental and ecological challenges in the design process.

  1. Line 301: “accelerative” should read “accelerate”;

Thank you. This has been corrected.

10) Figure 4: I wonder if the second theme in the “Happy Chicken” scenario should be “Climate Resilience” instead of “Changing Tastes”;

After internal discussions, the authors have decided to retain the original title due to its playful nature.

11)   Line 413: section 3.1.2 does not include any data on supply/delivery of resources/products; 

This should read ‘Baseline analysis of the typical poultry house captured carbon emissions associated with construction materials, heating, power, and supply’, as it refers to the supply of power only. See rows 1699-1700 for the updated text.

12)   Line 445: “… farms in in a…” should read “… farms in a…"

This has been updated. See row 1742.

13) Lines 576-584 & 629-636: Following previous points, I missed a more systematic approach and presentation of the different alternatives/scenarios addressed in this research as well as the relationships between queries and proposed solutions.

The authors provide one further example of how the compendium of muti-scalar ideas was developed, illustrated in Figure 4. This new figure is a matrix containing ideas for the house, farm and estate to reduce, reuse and produce resources. As stated in the response to comment 8, general and specific scenarios are visualised and discussed in more depth than in the original manuscript. Ultimately, the results in the proposition are not necessarily just about the scenarios developed in themselves. Instead, these propositions are used as vehicles to explore different design methods, such as designing with flows and prioritising different actors.

Round 2

Reviewer 1 Report

The authors have made an important effort in improving the manuscripts. Even when this reviewer may still have some different views on some points, he believes that the authors have made a good defense of those. What remains can be seen as part of the natural difference in interpretation. The article is much more solid now, deserving the opportunity of being presented to the scientific community. On that basis, this reviewer recommends publication in this present form.

Reviewer 2 Report

Dear authors,

thank you for the explanations provided to our comments.